# Prevalence of cancer related fatigue and its associated factors among adult cancer patients in eastern Ethiopia: A cross-sectional study

Henok Legesse[1]*, Yalew Mossie[1], Lidia Tolessa[1], Deribe Bekele Dechasa[1], Lencho Ahmedin[1], Seblewengel Fita[1], Aminu Mohammed Yasin[2], Seid Tesi[1], Ahmed Hiko[1], Sisay Habte[1], Addisu Alemu[3], Ahmed Mohammed[4], Michael Shawel[4]

**1** School of Nursing, College of Health and Medical Sciences, Haramaya University, Harar, Ethiopia, **2** Department of Midwifery, College of Medicine and Health Sciences, Dire Dawa University, Dire Dawa, Ethiopia, **3** School of Public Health, College of Health and Medical Sciences, Haramaya University, Harar, Ethiopia, **4** School of Medicine, College of Health and Medical Sciences, Haramaya University, Harar, Ethiopia

* henok_legesse@yahoo.com

## Abstract

### Background

Fatigue is a frequent and distressing symptom experienced by patients with cancer. It may result from the disease process and/or its aggressive treatment, which substantially impact the quality of life of cancer patients. Furthermore, there is a paucity of evidence in the Eastern part of Ethiopia. Hence this study aimed to assess the prevalence of cancer related fatigue and its associated factors among adult cancer patients, Eastern Ethiopia.

### Methods

Hospital based cross-sectional study was conducted from 1st May to 30th August, 2023 among 422 systematically selected cancer patients. Data were collected using structured, interviewer administered questionnaire. The outcome variable was evaluated using the Brief Fatigue Inventory (BFI). Binary logistic regression analyses were performed to examine the association between the explanatory variables and the outcome variable. Adjusted Odds Ratio (AOR) with 95% Confidence Interval (CI) at a P value less 0.05 was used to declare statistically significant association.

### Results

Out of 422, 382 individuals with various cancer types participated in the study with a response rate of 90.5%. The prevalence of cancer related fatigue was found to be 71.2% (95%CI: 65.7–75.5). Rural residence (AOR = 2.84, 95%CI: 1.25–6.43), female sex (AOR = 3.25, 95%CI: 1.49–7.0), private occupational status (AOR = 6.44, 95%CI: 2.42–17.12), never used coffee (AOR = 7.02, 95%CI: 2.37–20.75), inpatient

**Data availability statement:** All relevant data are within the paper and its Supporting Information files.

**Funding:** The author(s) received no specific funding for this work.

**Competing interests:** The authors have declared that no competing interests exist.

**Abbreviations:** BFI: Brief Fatigue Inventory; CRF: Cancer Related Fatigue; HFCSH: Hiwot Fana Comprehensive Specialized Hospital; NCCN: The National Cancer Center Network; WHO: World Health Organization.

admission (AOR = 4.68,95%CI: 2.21–9.88) and advanced (AOR = 6.21, 95%CI: 2.61–14.78) & unclassified cancer stages (AOR = 4.84, 95%CI: 1.42–16.57) were significantly associated with cancer related fatigue.

## Conclusions

Nearly three out of four cancer patients in Eastern Ethiopia experienced cancer related fatigue. The findings highlight the need for a supportive care service including psychosocial counseling and targeted intervention for high risk groups such as female, rural residents, inpatients, private workers and advanced cancer patients. Further research is warranted to explore the protective role of coffee consumption.

## Background

Fatigue is a frequent and distressing symptom experienced by patients with cancer. It is defined as a complex, subjective complaint characterized by persistent tiredness, lack of energy and weakness [1–3]. While fatigue, tiredness and exhaustion are often used interchangeably, a more comprehensive and widely accepted definition on Cancer Related Fatigue (CRF) is provided by the National Cancer Center Network (NCCN) as "a distressing, persistent, subjective sense of physical, emotional, and/or cognitive tiredness or exhaustion related to cancer or cancer treatment that is not proportional to recent activity and interferes with usual functioning" [4].Unlike fatigue or tiredness among healthy individuals, CRF is more severe and is persistent in spite of rest or relaxation, disproportionate to activity level and resists to conventional remedies including sleeping and lasts for weeks, months or years. A Brief Fatigue Inventory (BFI) scale score of 4 and above is considered to be clinically significant [5].

Cancer related fatigue (CRF) is troublesome complaint reported across various cancer types and treatment modalities. Cancer cells competes and consumes body's nutrients and calories that are needed for normal function, continuous cytokines release from immune inflammatory reactions and treatment modalities further intensifies the exhaustion [6].

Globally CRF is highly prevalent. A systematic review and meta-analysis with over 71,000 cancer patients estimated an overall prevalence of 49%, ranging from 26.2% in gynecological cancer patient to 60.6% among those with advanced stage disease [7]. Even though, the intensity and course of CRF differs among cancer cases and persisted for decades after completion of therapy in cancer survivors, the physical, emotional and mental exhaustion is burdensome [8]. Patients often perceive CRF as more distressing than other cancer related symptom including pain, nausea and vomiting [4].

Persistent CRF disrupts the patients' social role, work productivity and activities of daily livings which led to greater disability and poorer health related quality of life [4,9]. further a prospective study indicated that fatigue among cancer patients was significantly associated with enhanced mortality rate [9].

Despite its burden, CRF remains underreported, under-diagnosed and poorly managed partly due to healthcare providers often prioritize cancer treatment over symptom control, and because the multi-factorial determinants are not well understood [4,10]. Evidence suggested that CRF is influenced by a combination of physical, clinical, psychosocial factors [11]. A study conducted in Ethiopia reported that stage of cancer, presence of infection, type of cancer, and type of treatment had shown a significant association with CRF [12]. Effective management therefore requires and NCCN clinical practice guidelines suggests a multidisciplinary approach including patient/ family education, structured exercise program, self-monitoring of fatigue, energy conservation and psychosocial intervention, nutritional consultation and in selected cases pharmacological interventions such as psycho-stimulants and treating for pain, emotional distress and anemia as part of the integral management [4,10,13].In Ethiopia limited studies explored CRF. A study conducted in Addis Ababa reported a prevalence of 74.8% among cancer patients [12]. However, there is an information gap in the Eastern part of Ethiopia where differences in socio-demographic, clinical and behavioral conditions may influence both the prevalence and associated factors of CRF. Addressing this gap is crucial to inform tailored interventions and supportive care strategies. Therefore, the present study aimed to assess the prevalence of CRF and identify its associated factors among adult cancer patients in Eastern Ethiopia.

## Methods

### Study design and setting

The study was conducted in Hiwot Fana Comprehensive Specialized Hospital (HFCSH), Eastern Ethiopia from 1st May to 30th August, 2023. The hospital is located in Harar, the capital city of Harari Regional State. The city is located 526kms toward east of Addis Ababa, the capital city of Ethiopia. Hiwot Fana Comprehensive Specialized Hospital (HFCSH), a tertiary level teaching hospital, intended to serve an estimated of up to 5 million people in its surrounding catchment areas [14]. HFCSH cancer center which was established in 2020 is the only cancer center in the Eastern part of Ethiopia that provides inpatient and outpatient cancer care services.

### Study participants and sampling methods

All cancer patients who are adults (above 18 years) & clinically diagnosed and confirmed with pathology or cytology examination were included. Cancer patients below 18 years, newly diagnosed patients at first presentation or no prior follow up, known cognitive impairment [in understanding and replying to questions] or critically ill or incomplete patient records were excluded.

Sample size was determined by using single population proportion formula with the prevalence of CRF, P as 50%, 95% confidence interval (Z = 1.96) and 5% margin of error(d). In this consideration, the estimated sample size resulted 384. Adding 10% (n = 38) of non-respondent rate, the final sample size yielded 422.

$$n = (Z\alpha/2)2 * P (1 - P) / d2$$

$$= (1.96)2 * 0.5(1 - 0.5) / (0.05)2$$

$$= 384$$

The study was conducted in purposively selected HFCSH cancer center, since it is the only center which provides cancer related services in the catchment area. The study participants were recruited from the cancer registry on the basis of the eligibility criteria. The sampling frame was constructed by considering the average monthly 160 visits for the study period (N = 640). Then study participants was selected using systematic sampling technique (K = 1.5) (Fig 1).

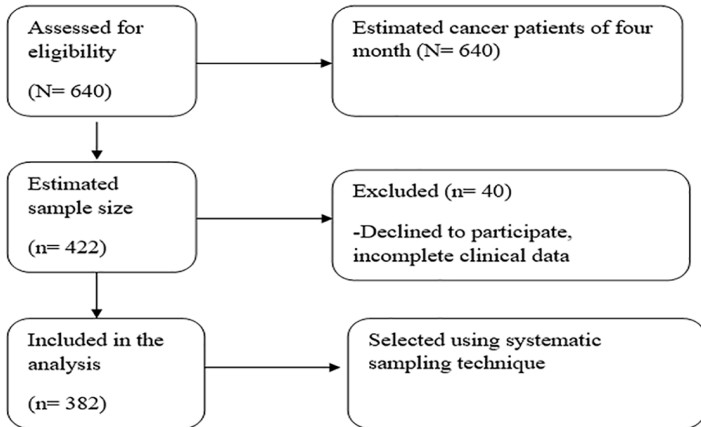

**Fig 1. CONSORT-style flow diagram for Cancer-Related Fatigue (CRF) among cancer patients in Eastern Ethiopia (n = 382).**

## Data collection tool and procedure

Data was collected using structured, self-reported questionnaire through an interview. The questionnaire was adapted from critically reviewing related literatures from different databases: including Pubmed, Google scholar, Hinari, web of science and CINHAL, using the key terms of "conceptual frame works", "associated factors" and "determinants" of CRF. The outcome variable was evaluated using The Brief Fatigue Inventory (BFI). The BFI instrument was first developed by Mendoza et al, and is known to have good psychometric property to measure CRF [15,16]. The BFI-Am, the Amharic version (i.e., the national language of Ethiopia), was investigated and validated to have an adequate psychometric property (an overall cronbach's alpha of 0.97) to use it in symptom research among Ethiopian cancer patients [17]. The overall questionnaire is translated to Amharic and Afan- Oromo by native language translators.

The questionnaire was divided in to three parts. The first section (part I) seek response for socio-demographic characteristics. The second section (part II) requests physical and behavioral factors. (Part III) looks for factors related to psychological characteristics. (Part IV) asks for disease and treatment factors. The last section (Part V) determines the level of fatigue using the Brief Fatigue Inventory (BFI) scale [S1 File]. The BFI contains nine items. The first three questions assess the "Now", "Usual", "Worst" level of fatigue in the past 24 hours. The responses are evaluated in numerical scale of 0–10, in which 0 implies "No fatigue" and 10 implies "fatigue as a bad as you can imagine". The rest six items assesses the interference of fatigue on general activity, mood, walking ability, normal work, relationships with other people and enjoyment of life in the past 24 hours. The responses are evaluated in numerical scale of 0–10, in which 0 implies "does not interfere" and 10 implies "completely interferes".

Data was collected by 3 Bsc nurses who are working in the oncology center of HFCSH. Brief orientation was given to data collectors on the study objectives, sampling, consent, data privacy, and checking for data clarity and completeness by the investigators. Then data collectors described the intent of the study and sought consent from the study participants. Finally, data was collected by using self-report questionnaire through an interview from the study participants. Clinical and disease related data was filled from the patient's medical card. To assure the quality of the data, pretest was done on 5% of the total estimated sample before the actual data collection on selected chronically ill patients (i.e., chronic follow up) for clarity and completeness in the same hospital. A two-day training was given to the data collectors and the supervisors. The investigators supervised the overall data collection process.

### Outcome variable measurements

Cancer Related Fatigue: The outcome variable in this study was CRF. Respondents rate the nine items of BFI on a numerical scale of 0–10. Then a global/sum score [sum of nine items divide by nine] was used to categorize the fatigue as Mild: 1–3, Moderate: 4–6 and Severe: 7–10. However, in this study a sum score of 4 and above (clinically significant fatigue) was used to describe the prevalence of CRF and to identify its associated factors [4,15,16].

**Social support:** A sum score of *Oslo social support scale (OSSS-3)* will be use. Social support is categorized as: 3–8 poor; 9–11 moderate and; 12–14 strong social support [18].

**Substance use** (smoking/coffee/alcohol/chat): "never substance user" was an individual who had never tried the substance in their lifetime. A "former user" was an individual who had used the substance in the past but stopped 30 days before the data collection period. A "current user" was an individual who used the substance at least one times in the past 30 days [19].

**Performance status:** A sum score of the Eastern Cooperative Oncology Group (ECOG) performance status scale was used. A grade of 0–1 indicates good performance status and a grade of 2–4 indicates poor performance status [20].

**Pain:** A sum score of *Brief pain inventory (BPI)* will be used. A score was categorized as mild: 1–3, moderate: 4–6 and severe: 7–10 [16,21]

**Unclassified cancer stage**: refers to patients without documented staging due to incomplete diagnostic workup or late presentation.

### Data processing and analysis

Collected data were entered into EpiData 3.1 and exported to SPSS (version 24) for cleaning and analysis. Frequency, proportion, mean, standard deviation, median and the inter-quartile range were computed. Bivariable and multivariable logistic regression analyses were performed to examine the association between the explanatory variables with CRF. Assumptions for logistic regression such as; multi-collinearity (correlation coefficient; VIF, tolerance) between the explanatory variables was checked (VIF was < 1.4). Hosmer-Lemeshow goodness-to-fit model was tested for multivariable fitness and the result was 0.12. Adjusted Odds Ratio (AOR) with 95% Confidence Interval (CI) at a P value less 0.05 was used to declare statistically significant association.

### Ethical approval and consent to participate

Ethical clearance for the study was received from Haramaya University, College of Health and Medical Science, Institutional Health Research Ethical review Committee/IHRERC (Reference No.: IHRER/017/2023). A formal letter was submitted for HFCSH hospital administrations ahead of conducting the data collection. Informed, voluntary, written and signed consent was sought from each participant using participant information sheet and consent form. The data collection procedure was performed in consideration with the declaration of Helsinki. And to not interfere with the patients' treatment schedule interviews was conducted after follow up visit completion. The interview took place in separated quite environment privately.

## Results

### Socio demographic characteristics of study participants

Out of 422 selected study participants, 382 individuals with cancer took part in this study, achieving a response rate of 90.5%. The mean age was 48 years (with SD of ±14.8) and almost half of them, 169 (44.2%), resided in rural areas. The mean monthly income was 5358 Ethiopian Birr (with SD of ±3919.2). Among the study participants, 239 (62.6%) were female. Majority of participants, 298 (78%) were married. A total of 151 (39.5%) participants had no formal education and 99 (25.9%) were housewives. Most of the participants health care expenses, 223 (58.4%) were self-pay. Almost one-third (130, 34%) had poor social support (Table 1).

**Table 1. Socio-demographic characteristics of cancer patients in Eastern Ethiopia (n = 382).**

| Variables | Category | Frequency(n) | Percentage |
|---|---|---|---|
| Address | Rural | 169 | 44.2 |
| | Urban | 213 | 55.8 |
| Sex | Male | 143 | 37.4 |
| | Female | 239 | 62.6 |
| Religion | Muslim | 225 | 58.9 |
| | Orthodox | 134 | 35.1 |
| | Protestant | 22 | 5.8 |
| | Other* | 1 | 0.3 |
| Marital Status | Single | 29 | 7.6 |
| | Married | 298 | 78 |
| | Divorced/Separated | 39 | 10.2 |
| | Widowed/Widower | 16 | 4.2 |
| Educational Level | No formal education | 151 | 39.5 |
| | Primary School | 66 | 17.3 |
| | Secondary School | 44 | 11.5 |
| | High School and above | 121 | 31.7 |
| Occupational Status | Private | 55 | 14.4 |
| | Public | 78 | 20.4 |
| | Farmer | 77 | 20.2 |
| | Merchant | 51 | 13.4 |
| | House wife | 99 | 25.9 |
| | Other** | 22 | 5.8 |
| Medical payment | Self-pay | 223 | 58.4 |
| | CBHI/Credit | 159 | 41.6 |
| Social support | Poor | 130 | 34 |
| | Moderate | 106 | 27.7 |
| | Strong | 146 | 38.2 |

*Catholic ** Daily laborer, student, self-employed.

### Clinical and disease related characteristics of study participants

Almost half of study participants (181, 47.4%) were inpatient admissions. A notable proportion were accounting for breast cancer (118, 30.9%) followed by gynecological (96, 25.1%) and gastrointestinal (96, 25.1%) cancers. The median duration since diagnosis was 7 months (IQR of ± 6). Furthermore, a substantial amount of the participants (162, 42.4%) were at stage II cancer. Notably, 265 (69.4%) of the participants fell into the category of chemotherapy treatment modality, 63 (16.5%) of the participants fell in to both chemotherapy and surgery and 14 (3.7%) of participants fell in to the category of radiation treatment modality. Almost all participants 318 (83.2%) and 354(92.7%) had no prior infection and co-morbidity respectively (Table 2).

### Prevalence of Cancer Related Fatigue

The prevalence of clinically significant cancer related fatigue was found to be 71.2% (95%CI: 65.7–75.5, n = 272). In which 101 (26.4%), 145 (37.9%) and 127 (33.2%) had mild, moderate and severe fatigue respectively (Fig 2).

**Table 2. Clinical and disease related characteristics of cancer patients in Eastern Ethiopia (n = 382).**

| Variables | Category | Frequency(n) | Percentage (%) |
|---|---|---|---|
| Types of admission | Out patient | 201 | 52.6 |
| | In patient | 181 | 47.4 |
| Cancer type | Breast | 118 | 30.9 |
| | Gynecological | 96 | 25.1 |
| | Gastrointestinal | 96 | 25.1 |
| | Hematological | 30 | 7.9 |
| | Hepato-biliary | 29 | 7.6 |
| | Other* | 13 | 3.4 |
| Stage of cancer | Stage I | 45 | 11.8 |
| | Stage II | 162 | 42.4 |
| | Stage III | 112 | 29.3 |
| | Stage IV | 36 | 9.4 |
| | Not classified | 27 | 7.1 |
| Current treatment condition | Remission induction | 235 | 61.5 |
| | Continuation | 112 | 29.3 |
| | Maintenance | 35 | 9.2 |
| Current treatment modalities | Radiation therapy | 14 | 3.7 |
| | Chemotherapy | 265 | 69.4 |
| | Surgery | 2 | 0.5 |
| | Chemo & radiation | 38 | 9..9 |
| | Chemo & surgery | 63 | 16.5 |
| Co-morbidity | Yes | 28 | 7.3 |
| | No | 354 | 92.7 |
| BMI | Under weight | 35 | 9.2 |
| | Normal weight | 300 | 78.5 |
| | Overweight | 42 | 11 |
| | Obesity | 5 | 1.3 |
| Performance status | Good performance | 245 | 64.1 |
| | Poor performance | 137 | 35.9 |
| BPI pain level | No pain | 18 | 4.7 |
| | Mild | 109 | 28.5 |
| | Moderate | 152 | 39.8 |
| | Severe | 103 | 27 |

*ENT, skin, male reproductive organ.

## Factors associated with cancer-related fatigue

In the bivariable analysis out of 26 variables, only 14 were significant at p- value <0.25. The results of the bivariable analysis indicated variables, including address, sex, marital status, occupation status, medical payment, BMI, coffee use, alcohol use, khat use, performance status, types of admission, cancer type, stage of cancer and current treatment modality were significant. These variables were then further analyzed through multivariable analysis, which revealed that only 6 variables- rural residence, female sex, private occupational status, people who never used coffee, inpatient admission and advanced & unclassified cancer stages- were significantly associated with cancer related fatigue at p value <0.05 (Table 3).

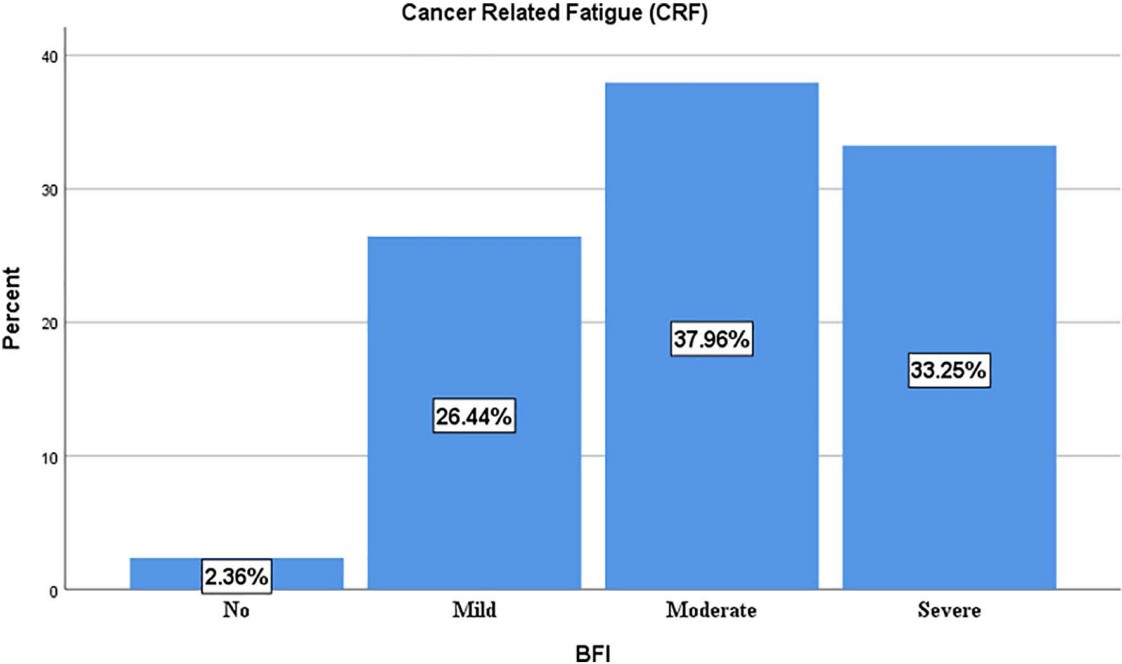

**Fig 2. The prevalence of Cancer Related Fatigue (CRF) by severity among cancer patients in Eastern Ethiopia (n = 382).**

In the conducted study, participants who came from rural areas were 2.84 times more likely to had cancer related fatigue as compared to participants who came from urban areas (AOR = 2.84,95%CI: 1.25–6.43).

Female participants were 3.25 times more likely to had cancer related fatigue as compared to male participants (AOR = 3.25, 95% CI: 1.49–7.08).

Those participants who had private occupation were 6.44 times more likely to had cancer related fatigue as compared to participants who had public and other occupational status (AOR = 6.44, 95% CI: 2.42–17.12).

Participants who never used coffee were 7.02 times more likely to had cancer related fatigue as compared to participants who used coffee occasionally/daily (AOR = 7.02, 95%CI: 2.37–20.75).

Participants who were at inpatient admission were 4.68 times more likely to report cancer related fatigue as compared to participants who were at the outpatient admission (AOR = 4.68,95%CI: 2.21–9.88).

As for the stage of cancer, those participants with advanced stages (III & IV) and unclassified stages were 6.21 and 4.84 times more likely to report cancer related fatigue as compared to participants with early stages (I & II) respectively (AOR = 6.21, 95%CI: 2.61–14.78) and (AOR = 4.84, 95%CI: 1.42–16.57).

## Discussion

This study was aimed to assess the prevalence of cancer related fatigue and its associated factors among adult cancer patients in Eastern Ethiopia. The prevalence of clinically significant cancer related fatigue was found to be 71.2% (95%CI: 65.7–75.5). Rural residence, female sex, private occupational status, never used coffee, inpatient admission and advanced & unclassified cancer stages were significantly associated with cancer related fatigue.

In this study the prevalence of CRF was found to be 71.2%, which is in line with a study conducted in Addis Abeba (74.8%) Ethiopia [12]. Nevertheless a slightly higher prevalence was reported from Amhara Region (77.3%) and Hawassa (77.4%), Ethiopia [22,23]. This might be due to the clinical and psychosocial characteristics variation in those

**Table 3. Bivariable and multi variable logistic regression analysis for factors associated with Cancer Related Fatigue (CRF) among cancer patients in Eastern Ethiopia (n = 382).**

| Variables | Category | CRF | | COR (95%CI) | AOR (95%CI) | P-value |
|---|---|---|---|---|---|---|
| | | No | Yes | | | |
| Address | Rural | 31 | 138 | 2.62(1.63-4.24) | 2.84 (1.25-6.43) | **0.013*** |
| | Urban | 79 | 134 | 1 | 1 | |
| Sex | Male | 48 | 95 | 1 | 1 | |
| | Female | 62 | 177 | 1.44 (0.92-2.26) | 3.25(1.49-7.08) | **0.003*** |
| Marital status | Unmarried | 17 | 67 | 1.78(0.99-3.21) | 2.34(0.91-5.9) | 0.076 |
| | Married | 93 | 205 | 1 | 1 | |
| Occupation | Private | 39 | 144 | 1.62(0.9-2.74) | 6.44(2.42-17.12) | **0.000*** |
| | Public | 34 | 44 | 0.57(0.31-1.03) | 1.01(0.34-2.99) | 0.995 |
| | Other | 37 | 84 | 1 | 1 | |
| Medical payment | Self-pay | 58 | 165 | 1.38(0.88-2.16) | 0.91(0.42-2.01) | 0.825 |
| | CBHI/Credit | 52 | 107 | 1 | 1 | |
| BMI | Under/Normal | 93 | 242 | 1.47(0.77-2.80) | 1.51(0.56-4.02) | 0.411 |
| | Overweight/Obese | 17 | 30 | 1 | 1 | |
| Coffee use | Never | 11 | 92 | 4.60(2.35-9.00) | 7.02(2.37-20.75) | **0.000*** |
| | Occasionally/Daily | 99 | 180 | 1 | 1 | |
| Alcohol use | Never/former | 101 | 224 | 0.30(0.03-2.45) | 0.21(0.02-2.20) | 0.196 |
| | Current | 9 | 48 | 1 | 1 | |
| Khat use | Never/Former | 83 | 210 | 1.10(0.65-1.85) | 0.78(0.28-2.13) | 0.624 |
| | Current | 27 | 62 | 1 | 1 | |
| Performance status | Good | 97 | 148 | 1 | 1 | |
| | Poor | 13 | 124 | 6.25(3.34-11.7) | 30.01(10.38-86.77) | **0.000*** |
| Type of admission | Outpatient | 77 | 124 | 1 | 1 | |
| | Inpatient | 33 | 148 | 2.78(1.73-4.46) | 4.68(2.21-9.88) | **0.000*** |
| Cancer type | Breast | 16 | 102 | 1.91(0.47-7.70) | 7.42(0.99-55.08) | 0.050 |
| | Gynecological | 32 | 64 | 0.60(0.15-2.33) | 0.54(0.08-3.69) | 0.526 |
| | Gastro-intestinal | 47 | 49 | 0.31(0.08-1.20) | 0.51(0.78-3.36) | 0.485 |
| | Hematology | 6 | 24 | 1.20(0.25-5.76) | 3.09(0.28-34.52) | 0.358 |
| | Hepato-biliary | 6 | 23 | 1.15(0.23-5.54) | 1.18(0.09-14.94) | 0.899 |
| | Other | 3 | 10 | 1 | 1 | |
| Stage of cancer | Stage I & II | 86 | 121 | 1 | 1 | |
| | Stage III & IV | 15 | 133 | 6.30(3.45-11.49) | 6.21(2.61-14.78) | **0.000*** |
| | Not classified | 9 | 18 | 4.43(1.69-11.60) | 4.84(1.42-16.57) | **0.012*** |
| Cancer treatment | Mono therapy | 90 | 191 | 0.52(0.30-0.90) | 0.47(0.19-1.16) | 0.102 |
| | Combination therapy | 20 | 81 | 1 | 1 | |

study participants. A higher proportion in advanced cancer stages, co-morbidity, chemotherapy treatment, poor social support was noted in those studies. In addition a higher finding was noted in Jordan (87.5%) [24]. This might be due to the socio-demographic variation, outcome measurement tools. The cancer type predominantly breast, lung and leukemia could be the possible reason for the higher fatigue level in Jordan. However, significantly lower findings were reported in Eastern China (52%) [25], Canada (40%,breast cancer) [26] and a systematic review and meta-analysis done in Netherlands (26.7%) [27], 49% [7]. This might be due to better health care infrastructures, early health seeking behaviors and better supportive care in those developed countries. In addition heterogeneity in populations and

outcome measurement tools could be a possible reason for the discrepancy in comparison with the systematic reviews and meta- analysis.

In the conducted study participants who came from rural areas were more likely to had cancer related fatigue as compared to participants who came from urban areas. This finding is in line with a study suggesting rural resident cancer patients had significantly poorer physical quality of life as manifested in fatigue [28,29]. This might be due to strenuous and demanding job, poor health seeking behavior and limited comprehensive health service access due to long travel distance and transportation [30,31].

Female participants were more likely to had cancer related fatigue as compared to male participants. This is in con- gruent with a systematic review and meta-analysis studies [7,32]. This could be possibly due to the biological difference, symptom perception and psycho-social distress including anxiety, depression and emotional burden prevalent among females [33–35].

Those participants who had private occupation had more likely cancer related fatigue as compared to participants who had public and other occupational status. This might be due to job insecurity and limited workplace accommodation inter- feres with treatment coupled with emotional distress worsen fatigue severity [36,37].

Participants who never used coffee were more likely to had cancer related fatigue as compared to participants who used coffee occasionally/daily. This finding is indirectly supported by a study conducted to determine anti-fatigue and anti-tumor effect of coffee in tumor bearing mice [38]. This might be due to the caffeine effect through enhanced lipolysis and the restored Non-Esterified Fatty Acid (NEFA) levels could be used as an alternative energy source [38]. This could also be likely due to the psycho-stimulant mood improvement and enhanced appetite effect for better nutrition [39,40]. Further research is warranted to explore the protective role of coffee consumption.

Participants who were inpatient admissions were more likely to report cancer related fatigue as compared to participants who were at the outpatient admission. This is in line with a study conducted in Amhara region, Ethiopia [41]. This might be due to the complex care need and symptom burdens along with the psychological distress during hospitalization [42].

As for the stage of cancer, those participants with advanced stages (III & IV) and unclassified stages were more likely to report cancer related fatigue as compared to participants with early stages (I & II). This is in line with a systematic review and meta-analysis study [43] and a cross-sectional study done in Ethiopia [12,23,41]. This might be due to the increased tumor burden, systemic inflammation and aggressive treatment modality and toxicity [6].

## Limitations/weaknesses of the study

This study is limited by its cross-sectional design, which precludes causality. Self-reported data may introduce recall bias. The single-center setting may limit generalizability. The smaller sample size and the wider confidence interval of poor performance status limit the strength of association with CRF.

Coffee consumption was assessed crudely (ever/never); frequency and amount were not evaluated.

## Conclusions

Nearly three of four cancer patients participated had clinically significant cancer related fatigue in Eastern Ethiopia. Rural residence, female sex, private occupational status, never used coffee, inpatient admission and advanced & unclassified cancer stages were significantly associated with cancer related fatigue. An integrated approach targeting on supportive care service, psychosocial and financial counseling, and moderate use of coffee is recommended. High risk groups such as female, rural residents, inpatients, private workers might require tailored interventions.

## Supporting information

**S1 File. Questionnaire (English version).**
(DOCX)

## Acknowledgments

We would like to thank HFCSH administration for their relentless support. We would like to extend our gratitude to the study participants and data collectors without whom this study may not come to reality.

## Author contributions

**Conceptualization:** Henok Legesse, Yalew Mossie, Lidia Tolessa, Deribe Bekele Dechasa, Michael Shawel.

**Data curation:** Henok Legesse, Yalew Mossie.

**Formal analysis:** Henok Legesse, Yalew Mossie, Lencho Ahmedin, Seblewengel Fita, Aminu Mohammed Yasin, Seid Tesi, Ahmed Hiko, Sisay Habte, Addisu Alemu, Ahmed Mohammed.

**Investigation:** Henok Legesse, Yalew Mossie, Lidia Tolessa, Deribe Bekele Dechasa, Lencho Ahmedin, Seblewengel Fita, Aminu Mohammed Yasin, Seid Tesi, Ahmed Hiko, Sisay Habte, Addisu Alemu, Ahmed Mohammed.

**Methodology:** Henok Legesse, Yalew Mossie, Lidia Tolessa, Deribe Bekele Dechasa, Michael Shawel.

**Software:** Henok Legesse, Yalew Mossie, Lencho Ahmedin, Seblewengel Fita, Aminu Mohammed Yasin, Seid Tesi, Ahmed Hiko, Sisay Habte, Addisu Alemu, Ahmed Mohammed, Michael Shawel.

**Supervision:** Henok Legesse, Yalew Mossie, Deribe Bekele Dechasa, Lencho Ahmedin, Aminu Mohammed Yasin, Seid Tesi, Ahmed Hiko, Sisay Habte, Addisu Alemu, Ahmed Mohammed, Michael Shawel.

**Validation:** Henok Legesse, Yalew Mossie, Michael Shawel.

**Visualization:** Seblewengel Fita.

**Writing – original draft:** Henok Legesse, Yalew Mossie, Lidia Tolessa, Deribe Bekele Dechasa.

**Writing – review & editing:** Henok Legesse, Yalew Mossie, Lidia Tolessa, Deribe Bekele Dechasa, Lencho Ahmedin, Seblewengel Fita, Aminu Mohammed Yasin, Seid Tesi, Ahmed Hiko, Sisay Habte, Addisu Alemu, Ahmed Mohammed, Michael Shawel.

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
