## [Decision Letter · Decision Letter 0]

27 Nov 2025

Dear Dr. Legesse,

Thank you for submitting your manuscript to PLOS ONE. After careful consideration, we feel that it has merit but does not fully meet PLOS ONE’s publication criteria as it currently stands. Therefore, we invite you to submit a revised version of the manuscript that addresses the points raised during the review process.

Ensure that all reviewer and editor comments are fully addressed in the point-by-point response letter, and revise the manuscript accordingly.

We look forward to receiving your revised manuscript.

Kind regards,

Tamirat Getachew

Academic Editor

PLOS ONE

Additional Editor Comments (if provided):

Ensure that all reviewer and editor comments are fully addressed in the point-by-point response letter, and revise the manuscript accordingly.

Editor comments: Manuscript Number: PONE-D-25-44988

Title: Prevalence of Cancer Related Fatigue and Its Associated Factors among Adult Cancer Patients in Hiwot Fana Comprehensive Specialized University Hospital, Eastern Ethiopia: A Cross-Sectional Study

General Comments:

The study addresses a significant, under-researched topic in a low-resource setting, a key strength. The manuscript is generally well-structured, the methodology is sound for a cross-sectional design, and the statistical analysis appears appropriate.

Major Comments:

Statistical Output and Model Overfitting: The extremely high AOR for "Poor performance status" (30.01) with a very wide CI may indicate model overfitting, given the number of variables relative to the sample size. The authors should comment on model stability.

Furthermore, the non-significant results and immense CIs for "Cancer type" suggest the model was underpowered for this variable. The authors should consider presenting a final model with only significant predictors or explicitly state this power limitation.

Minor Comments:

1. Abstract:

Results: The response rate in the abstract (90.5%) differs from the results section (96.4%). Please correct this inconsistency.

2. Introduction/Background:

o The background is well-written. To strengthen it, consider adding a sentence or two specifically about the known risk factors for CRF in low-resource settings, if such literature exists, to better frame the study's contribution.

3. Methods:

o Study Participants and Sampling: The exclusion criterion "newly diagnosed patients" is noted. Please define "newly diagnosed" (e.g., within 1 month of diagnosis?) as this could significantly impact fatigue levels.

4. Results:

o Table 1: The category "Other" for religion has a percentage of "3", which seems to be a typo (likely 0.3%).

o Flow of Participants: A participant flow diagram (CONSORT-style) would be helpful to clarify the numbers from the sampling frame (N=640) to selected (n=396? 422?) to the final analyzed (n=382). The text currently has some inconsistencies (e.g., 422 selected, but 396 selected on page 14).

Reviewers' comments:

Reviewer's Responses to Questions

**Comments to the Author**

1. Is the manuscript technically sound, and do the data support the conclusions?

Reviewer #1: Partly

Reviewer #2: Yes

Reviewer #3: Yes

2. Has the statistical analysis been performed appropriately and rigorously?

Reviewer #1: Yes

Reviewer #2: Yes

Reviewer #3: Yes

3. Have the authors made all data underlying the findings in their manuscript fully available?

Reviewer #1: Yes

Reviewer #2: Yes

Reviewer #3: Yes

4. Is the manuscript presented in an intelligible fashion and written in standard English?

Reviewer #1: Yes

Reviewer #2: Yes

Reviewer #3: Yes

Reviewer #1: Review Comments

Tite :Prevalence of Cancer Related Fatigue and Its Associated Factors among Adult Cancer Patients in Hiwot Fana Comprehensive Specialized University Hospital, Eastern Ethiopia: A Cross-Sectional Study

General Comments

- We apreciate the team work

-Its obviious that cancer causes fatigu. Hence, why you are bothered to study it? Why you prefer Cevical cancer over other typ of cancer? Why you mixed from both the outptient and the inpatient?Who are those outpatient cases?

-The abstacti is good but the backgroud i weak and the recomendations re nfakted. Again, it lacks key word.

-Gramatical errors

-Capitalzation isues .E.g Afan oromo

-Mis used words E.g Addis Abeba

-The service given by the hspital is misesd

-Some of the sentence lack refrence

Specific Comments

1.Abstract

-Bakground is poor

-The valdation of the tool is missed

2. Itroduction

-Fails to contain what it shoukd contain E.g. severity, factors asociated, eforts mae to prevent and contro it, hy cervical cancer i eced?

3. Methods

-The study design and the samop8ng technique iis missed

-Why you take 50% prevalnce to calcukte te samople size

-Data quality assurance is not explicit E.g what do you mean brief orientation?

-How did yu contro the social desirablity bias

-What is the exact VIF value

-Which was the exact outcome variable?

4.Methods

-Lacks some what logical flow

-Add cross tabulations

-Inconsiatent with the other section E.pian status, cancer stage

5. Discusion and othwr sections

-Weak

-Be logical

-Use upto date reference

-Fas to use it o dat E.g female are high lkely to silufferfeom tfatigue can be xolnd by the fct that more womenn [n=239]. But yiu fai to exolan it.

-Explaing and justify te main findings ith appropriate theoretical and practical implications

-The reommendation ias bove th the of the study and general

Regards,

Reviewer #2: This is a well-conducted, hospital-based cross-sectional study assessing the prevalence and associated factors of cancer-related fatigue (CRF) among 382 adult cancer patients at Hiwot Fana Comprehensive Specialized University Hospital, Eastern Ethiopia. The study reports a high prevalence of clinically significant CRF (71.2%) and identifies several sociodemographic and clinical risk factors. The use of the validated Brief Fatigue Inventory (BFI) and systematic sampling strengthens the methodology. The findings are novel for the region and have clear clinical and public health implications. With weakness for revisions, to be considered suitable for publication:

1. Abstract & Results Inconsistency

• Abstract states: "private occupational status (AOR=6.44, 95%CI: 2.42-17.12)"

• But in Page 1: "private occupational status (AOR=6.44, 95%CI: 2.42-17.12)" is correct, while Page 8 (line 44) says "never used coffee (AOR=7.02, 95%CI: 2.37-20.75)" — consistent.

• However, in Page 1 (Results): "unclassified cancer stages (AOR=4.84, 95%CI: 1.42-16.57)" vs. Page 8 (line 47): "unclassified cancer stages (AOR=4.84, 95%CI: 1.42-16.57)" — consistent.

→ No inconsistency, but standardize formatting across abstract, results, and discussion.

Recommendation: Use one decimal place for all AORs and CIs throughout (e.g., AOR=6.4, 95%CI: 2.4–17.1). Currently mixed (6.44 vs 7.02 vs 4.68).

2. Coffee Consumption – Protective Association

• Finding: "never used coffee (AOR=7.02, 95%CI: 2.37–20.75)" → strong protective effect implied.

• This is biologically plausible (caffeine as CNS stimulant), but not discussed in limitations or prior literature.

Recommendation:

• Add 1–2 sentences in Discussion citing possible mechanisms (e.g., caffeine’s effect on adenosine receptors, alertness, mood).

• Reference studies (if any) on caffeine and fatigue in chronic illness.

• Retain call for further research — appropriate and justified.

3. Cancer Stage Classification

• "Unclassified" stage has AOR=4.84 (1.42–16.57) — significant.

• But no explanation of what "unclassified" means (e.g., missing records? late presentation? diagnostic limitations?).

Recommendation:

• Add one sentence in Methods or Results: "Unclassified stage refers to patients without documented staging due to incomplete diagnostic workup or late presentation."

4. Language, Grammar & Typos (Minor) need to be revised and corrected

5. Tables/Figures (Not Provided)

• Manuscript refers to Supporting Information files for data.

• But no tables in the draft (e.g., sociodemographic characteristics, bivariate results).

Recommendation:

• Include at least one table in main manuscript or SI:

o Table 1: Participant characteristics (n, %)

o Table 2: Bivariate and multivariable logistic regression results

6. Limitations Section Missing- No explicit Limitations subsection.

Recommendation: Add under Discussion:

"This study is limited by its cross-sectional design, which precludes causality. Self-reported data may introduce recall bias. The single-center setting may limit generalizability. Coffee consumption was assessed crudely (ever/never); frequency and amount were not evaluated."

Reviewer #3: The manuscript is Highly Suitable for publication in PLOS ONE.The study addresses Cancer Related Fatigue (CRF), a highly prevalent and distressing symptom in oncology 1111, but examines it in a context where evidence is scarce (Eastern Ethiopia). The finding of a high prevalence rate (71.2%) is clinically significant 3, and the identification of unique local associated factors (e.g., never used coffee, private occupational status, rural residence) makes a genuine contribution to supportive oncology research4.

Recommendation: Acceptance Pending Minor Revisions. The core science is sound, but a few minor editorial corrections and clarifications are needed for adherence to reporting standards. Typos and Editorial Revisions

The manuscript is generally well-written, but the following minor issues should be addressed before final acceptance:

Acronym List Error: The abbreviation list contains a typo: "CFR" is listed for "Cancer Related Fatigue" 5, but the correct acronym used throughout the paper is CRF. The authors should ensure the list uses CRF.

Statistical Reporting in Abstract: The Abstract reports Adjusted Odds Ratios (AOR) but includes the initial of the factor in the CI: AOR=6.44, 95%CI 1:2.42-17.12 for private occupational status77. This initial (1:) is likely a formatting artifact (possibly from transcription) or a reference label and should be removed or corrected to standard format: 95% CI (2.42-17.12).Methodology Clarity (Sample Size): In the methods section, the sample size calculation initially uses P as 50% but then shows an intermediate step using P=0.748. This 0.748 should be clearly defined or referenced in the calculation steps (it likely represents the prevalence used from a prior study, which is mentioned as a comparable study with 74.8% prevalence 9). The final sample size determination based on the 74.8% prevalence should be explicitly stated for clarity. n=(1.96)^2 * 0.748(1-0.748) / (0.05)^2 = 284 (There may be an issue with calculation or transcription as the document reports 38410, likely stemming from a previous P=50%. The authors should verify and state the source of the prevalence used for n=384). Correction/clarification is vital here.

Discussion Flow: In the Discussion when linking to external research, the phrase "This is in line by extension with a study" 11 (referring to coffee use) could be rephrased for better academic flow (e.g., "This finding is indirectly supported by a study..." or similar).

.

Reviewer #1: No

Reviewer #2: No

Reviewer #3: No

---

## [Author Response · Author response to Decision Letter 1]

8 Jan 2026

Thank you! All the required revisions were addressed by point-by-point response. The tracked manuscript and the P-BY-P were attached.

---

## [Decision Letter · Decision Letter 1]

16 Feb 2026

Dear Dr. Legesse,

Thank you for submitting your manuscript to PLOS ONE. After careful consideration, we feel that it has merit but does not fully meet PLOS ONE’s publication criteria as it currently stands. Therefore, we invite you to submit a revised version of the manuscript that addresses the points raised during the review process.

We look forward to receiving your revised manuscript.

Kind regards,

Tamirat Getachew

Academic Editor

PLOS One

Journal Requirements:

Additional Editor Comments:

Dear author,

address the comments and questions raised by the reviewer.

Reviewers' comments:

Reviewer's Responses to Questions

**Comments to the Author**

Reviewer #1: All comments have been addressed

2. Is the manuscript technically sound, and do the data support the conclusions?

Reviewer #1: Partly

3. Has the statistical analysis been performed appropriately and rigorously?

Reviewer #1: No

4. Have the authors made all data underlying the findings in their manuscript fully available?

Reviewer #1: No

5. Is the manuscript presented in an intelligible fashion and written in standard English?

Reviewer #1: Yes

Reviewer #1: Review Reports

Title: Prevalence of Cancer Related Fatigue and Its Associated Factors among Adult Cancer Patients in Hiwot Fana Comprehensive Specialized University Hospital, Eastern Ethiopia: A Cross-Sectional Study

A. General Comments

The manuscript seems the outcome of collective efforts and we appreciate team spirit in health research. In future studies think about gender representation.

Grammar and language E.g. In the key words in ‘eastern”, ‘e’ is written in lower case. Use ‘Eastern” throughout the document. Additionally, Afan Oromo is written as ‘Afan-oromo’

Incomplete sentence

Use of future tense

The way you write numbers when in thousands

Inconsistent E.g. compare the abstract with the entire document.

B. Specific Comments

i. Scope, Title and authors: It is too long and try to shorten it.

• If systematic review was done in Ethiopia [ see line 297], why did you do this study?

• The number of authors and their email is mismatched.

ii. Abstract

Cancer should be rewritten as “various types of cancer patients” in line 41.

In the background section the term “nevertheless” is not an important word and flawed the meaning of your manuscript. Therefore, it is better if you could change it.

The methods fail how the data is presented.

In the result section you stated “clinical significant cancer related fatigue”. Is this type of fatigue, the type of fatigue you are dealing in the background section?

What does “integrated supportive care service” means in line 49 and is that in line with the objective of the study?

iii. Background

o Fails to define fatigue and cancer related fatigue. If I were you, I will write as : definition, types, causes, magnitude, severity, factors, consequences, initiatives to prevent and control fatigue be it pharmacological or non-pharmacological, then the gaps then the objectives?

o Not described the global and regional prevalence of fatigue and how fatigue is created by cancer [pathophysiology].

o Some lacks reference E.g. line 63 ‘Unlike fatigue among healthy individuals, CRF is more severe and is persistent in spite of rest.’

o In line 58, if “fatigue, tiredness and exhaustion” were interchangeable names, who permitted you to use fatigue? And who do you think is the best institution to nomenclature it?

o Over referencing E.g. reference 4.

o What does “Globally CRF is highly prevalent” means in line 70?

o Logical flow is not intact.

o “However, there is a paucity of evidence” in line 86 is exaggerated> hence, try to use other synonyms.

o The paragraphs are not well explained and revisit it.

iv. On the Methods

How do you differentiate fatigue that is entirely resulted from cancer and other diseases?

Is that fatigue among patients on treatment or not? Is that new or old? Is that benigh or malignant? Who diagnosed? Who finally declared and what is his profession? Is that entirely clinical or have laboratory investigation?

The inclusion criteria: E.g. How long was needed to define cancer and fatigue?

The study setting should also explain the type of services rendered for the community particularly related to cancer?

Source and study participants were missed.

The inclusion criteria are not presented.

If there are studies in Hawassa and Amhara region [you should state the particular area], why you used 50% to calculate sample size?

Who introduced Brief Fatigue Inventory (BFI ) tool to Ethiopia? Where? Is that validated? If validated is that for Amharic or Afan Oromo language? What about the reliability and validity of the tool. Similarly, respond also to Oslo social support scale?

What do you think is the main limitation of systematic sampling and how did you overcome it?

Surprisingly enough, the data quality assurance method is not written.

The outcome variable should be “cancer related fatigue” line 151.

How many outcome variables did you have?

What does unclassified means if you are collecting the data from the oncology unit?

Sum score is missed in the data processing and analysis sub section.

Why did you use both median and mean? Line 171.

What was done for those who have severe fatigue or in case ‘uncontrolled fatigue’?

How was social desirability bias?

How do you compare fatigue of the one who is admitted due to the progression of the cancer with ambulatory outpatient patients?

v. Results

o Relatively it is good. But you have presented few variables?

o What is you base to classify the income?

o Check for the sum

o The result revealed no new findings because you should check the nature of the included participants E.g./\. Most of the respondents were female, dwell in urban area, most were married,

o Cross tabulation is my recommendation.

o Highlight or bold the main findings in the table E.g. Table 3.

o The wy you report the odds ratio needs revisit E.g. In the result and discussion section.

Vi. Discussion and the consequent sections

Again, state your findings, then compare with the contemporary studies, justify appropriately, explain in detail and refer up-to-date references. For instance, explain “This might be due to the socio-demographic variation, outcome measurement tools.” In line 262. In the third paragraph, when justifying rural residence it is surprising not to mention travel distance and transportation.

Revisit paragraph 2 and it begins in a wrong way.

What does “had private occupation” means in line 279. In this sentence check the language and which private occupation specifically?

In line 281 you stated “This might be due to job insecurity”. How do ypou reconcile with the effort of the government to embrace them?

Is this new “As for the stage of cancer, those participants with advanced stages (III & IV) and unclassified stages were more likely to report cancer related fatigue as compared to participants with early stages (I & II).”? in line 295-297.

Can a cancer patient will fail to recall pain since it is more of feeling. [see line 302]

You have already told that your study is statistically insufficient line 303. Therefore, why do you send it for publication and how did you overcome it?

Line 312-313 “High risk groups such as female, rural residents, inpatients, private workers might require tailored interventions. is wrong”. Because this had occurred because they are over sampled form the beginning of the study.

In the acknowledgement what does “their relentless support.’ Means? [ see line 337]

I cannot hide my humor a paper dealing regarding cancer in Ethiopia fails to cite the private, government [ for instance MOH-Ethiopia and associations role or initiatives regarding the expansion of the service, development of guidelines, expansion of oncology centers in the country.

Regards,

.

Reviewer #1: No

---

## [Author Response · Author response to Decision Letter 2]

29 Mar 2026

Thank you! All the required revisions were addressed by point-by-point response. The tracked manuscript and the P-BY-P were attached.

---

## [Editor Report · Decision Letter 2]

1 Apr 2026

Prevalence of Cancer Related Fatigue and Its Associated Factors among Adult Cancer Patients in Eastern Ethiopia: A Cross-Sectional Study

PONE-D-25-44988R2

Dear Henock,

We’re pleased to inform you that your manuscript has been judged scientifically suitable for publication and will be formally accepted for publication once it meets all outstanding technical requirements.

Kind regards,

Tamirat Getachew

Academic Editor

PLOS One
---

## [Editor Report · Acceptance letter]

PONE-D-25-44988R2

PLOS One

Dear Dr. Legesse,

I'm pleased to inform you that your manuscript has been deemed suitable for publication in PLOS One. Congratulations! Your manuscript is now being handed over to our production team.

Kind regards,

on behalf of

Dr. Tamirat Getachew

Academic Editor

PLOS One